# Molecular Implications of Natriuretic Peptides in the Protection from Hypertension and Target Organ Damage Development

**DOI:** 10.3390/ijms20040798

**Published:** 2019-02-13

**Authors:** Speranza Rubattu, Maurizio Forte, Simona Marchitti, Massimo Volpe

**Affiliations:** 1Department of Clinical and Molecular Medicine, School of Medicine and Psychology, Sapienza University of Rome, 00189 Rome, Italy; massimo.volpe@uniroma1.it; 2IRCCS Neuromed, 86077 Pozzilli, Italy; maurizio.forte@neuromed.it (M.F.); simona.marchitti@neuromed.it (S.M.)

**Keywords:** natriuretic peptides, hypertension, stroke, cardiac hypertrophy, linkage analysis, genetic variants, animal models

## Abstract

The pathogenesis of hypertension, as a multifactorial trait, is complex. High blood pressure levels, in turn, concur with the development of cardiovascular damage. Abnormalities of several neurohormonal mechanisms controlling blood pressure homeostasis and cardiovascular remodeling can contribute to these pathological conditions. The natriuretic peptide (NP) family (including ANP (atrial natriuretic peptide), BNP (brain natriuretic peptide), and CNP (C-type natriuretic peptide)), the NP receptors (NPRA, NPRB, and NPRC), and the related protease convertases (furin, corin, and PCSK6) constitute the NP system and represent relevant protective mechanisms toward the development of hypertension and associated conditions, such as atherosclerosis, stroke, myocardial infarction, heart failure, and renal injury. Initially, several experimental studies performed in different animal models demonstrated a key role of the NP system in the development of hypertension. Importantly, these studies provided relevant insights for a better comprehension of the pathogenesis of hypertension and related cardiovascular phenotypes in humans. Thus, investigation of the role of NPs in hypertension offers an excellent example in translational medicine. In this review article, we will summarize the most compelling evidence regarding the molecular mechanisms underlying the physiological and pathological impact of NPs on blood pressure regulation and on hypertension development. We will also discuss the protective effect of NPs toward the increased susceptibility to hypertensive target organ damage.

## 1. Introduction

Hypertension is a complex trait that results from both environmental and genetic factors [1]. Several mechanisms have been highlighted as potential contributors to the etiopathogenesis of essential hypertension, mostly based on common knowledge of blood pressure regulation. In this regard, the biological properties of the cardiac natriuretic peptides (NPs)—that is, natriuresis, diuresis, and vasorelaxation [2], implying a direct modulatory role on blood pressure homeostasis—make all components of the NP family, including the natriuretic peptide receptors (NPRA (type A natriuretic peptide receptor), NPRB (type B natriuretic peptide receptor), and NPRC (type C natriuretic peptide receptor)), and the related protease convertases (furin, corin, PCSK6), as major etiopathogenetic candidates for hypertension. Moreover, the functional properties of these hormones involve anti-hypertrophic, anti-proliferative, and anti-inflammatory effects [3] that may significantly impact on the process of cardiovascular remodeling, such as that observed in hypertension.

Herein, we present evidence supporting the contribution of NPs to hypertension development obtained by both experimental and clinical investigations. As will be outlined below, the most compelling evidence derives from the genetic approach. The impact of NPs on the pathogenesis of hypertensive target organ damage will be also discussed.

## 2. Role of NP Circulating Levels in the Development of High Blood Pressure

Several physiological experiments have shown that αANP, the product of the cleavage of proANP by corin, along with the N-terminal residues 1–98, is a key regulator of systemic blood pressure through its natriuretic, diuretic, and vasorelaxant properties, all mediated by the type A natriuretic peptide receptor (NPRA) [3]. As a consequence, a reduction of ANP favors high blood pressure, whereas its increase may serve as a pharmacological tool to lower blood pressure levels. In fact, evidence that ANP levels contribute to hypertension has been initially shown by testing the effects of its infusion in hypertensive patients [4,5]. Subsequently, the anti-hypertensive effect of NPs has been documented in different pathological contexts, such as primary aldosteronism [6], pheochromocytoma [7], and hyperthyroidism [8]. In each of these conditions, the increase of NPs is able to counteract the pro-hypertensive effect of the specific endocrine disorder. Moreover, since diuresis represents a major action of NPs, the diuretic therapy reduces NP levels [9].

The genetic/molecular approach, undertaken over the last three decades, has the merit to provide further support to the significant association between NP levels and hypertension. In fact, it was shown that the knockout of *Nppa* (the gene encoding ANP) led to salt-sensitive hypertension in mice [10]. Consistently, the overexpression of *Nppa* led to hypotension [11]. Similarly, lack of NPRA caused salt-sensitive hypertension in mice [12].

On the other hand, the biologically active carboxy-terminal peptide (BNP 1–22), derived from the cleavage of the proBNP precursor by both furin and corin [13], appears to have a weaker impact on the pathogenesis of hypertension compared to ANP at the experimental level. In fact, deletion of *Nppb* in mice led to cardiac fibrosis rather than to hypertension development [14]. A hypertensive effect due to lack of *Nppb* was documented only in a rat model [15].

The results of the genetic manipulations in rodents stimulated several studies aimed at identifying the contribution of NPs to human hypertension. In this regard, the association of human *NPPA/NPPB* gene variations with circulating ANP and BNP levels was investigated for selected single nucleotide polymorphisms (SNPs) with the achievement of some remarkable results. Of note, we reported an association of the –664C > G minor allele located within the NPPA promoter and associated with lower plasma ANP levels, early onset of blood pressure increase, and the predisposition to develop hypertension in a general population from Southern Italy [16]. Contrasting evidence was obtained for the same SNP in a Japanese cohort of hypertensive patients [17].

Another NPPA variant, rs5063 (–664G > A), falling within the exon 1 of the gene and responsible of a Val-to-Met transition, was associated with blood pressure progression in the Women’s Genome Health Study and with reduced blood pressure levels in a Chinese population [18,19]. A SNP in linkage disequilibrium with rs5063 (1837G > A, detected by a *Hpa*II restriction fragment length polymorphism) was associated with higher occurrence of hypertension in whites and African–American subjects [20]. Interestingly, two studies associated variants near the NPPA/NPPB locus with proANP, BNP, and NT-proBNP levels [21,22]. Among others, the minor allele at rs5068 NPPA variant turned out to be associated with higher ANP levels and with hypertension risk [21]. In this regard it has been shown that this variant falling within the 3’-UTR of NPPA, affects the quantity of the ANP transcript through an interference with the microRNA-425 [23]. In particular, in the presence of rs5068 minor allele, the microRNA-425 cannot bind to NPPA to inhibit the gene transcription. As a consequence, the ANP levels increase [23]. Moreover, lower blood pressure levels and reduced risk of hypertension were observed [21,24,25,26].

Subsequently four genome-wide association studies (GWAS) analyzed circulating BNP or NT-proBNP levels in association with *trans* loci near LOXL2, SLC39A8, KLKB1, and GALNT4 [27,28,29,30]. As a result of the abovementioned studies, genetic variants at the MTHFR-NPPB locus (mapping on human chromosome 1 and containing both NPPA and NPPB) appeared to act through increased ANP/BNP production to lower blood pressure levels and, consequently, to influence susceptibility to hypertension development. However, there was a need to more precisely identify the variants truly associated with a change in NP levels within the MTHFR-NPPB locus and, therefore, responsible for the hypertensive effects, which prompted subsequent investigations. In fact, a recent study testing eight independent genetic variants in two known loci (NPPA-NPPB and POC1B-GALNT4) and one novel locus (PPP3CC) found that only those variants correlated with midregional proANP levels had a statistically significant, albeit weak, impact on blood pressure, whereas variants affecting BNP levels did not [31].

Although the latter evidence appeared to further support the experimental findings in favor of a major role of ANP, rather than BNP, on blood pressure regulation and hypertension development, NPPB cannot be completely ruled out as a “hypertensive” gene. A single SNP, the rs198389 functional variant in the NPPB promoter region, is associated with NT-proBNP levels in several populations [32]. In a large biracial prospective cohort study, the rs198389 NPPB promoter variant was found to be highly associated with large differences in NT-proBNP levels in both black and white populations. Patients with the AG and GG genotypes had progressively higher NT-proBNP levels compared to those with AA genotype. Patients with the GG genotype had reduced systolic blood pressure and diastolic blood pressure levels and were 15% less likely to take anti-hypertensive medications and 19% less likely to have a diagnosis of hypertension [33].

## 3. Role of Other Components of the NP Family

The involvement of other members of the NP family in protection from the development of hypertension has been mainly discovered through the genetic approach, starting with the experimental evidence in mice and then moving to the human disease. Thus, both gene deletions in mice and functional variants of the corresponding human genes encoding corin, furin, NPRA, and NPRC receptors have been associated with hypertension.

Corin is the physiological proANP convertase that activates proANP in a sequence-specific manner [34]. Blocking corin expression inhibits proANP processing in cardiomyocytes [35]. The fundamental relevance of corin for the maintenance of normal blood pressure levels was revealed by the corin knockout mice model. This model carries undetectable levels of mature ANP and it develops salt-sensitive hypertension with cardiac hypertrophy [36].

A recent study revealed that proprotein convertase subtilisin/kexin-6 (PCSK6), a protease belonging to the PCSK family, is the long sought-after specific corin-activating enzyme and, as such, is a critical regulator of the whole cascade leading to proANP processing and to αANP release into circulation, ultimately controlling water–electrolyte and blood pressure homeostasis [37,38]. PCSK6 cleaves corin at the conserved activation site, converting zymogen corin to an active enzyme. PCSK6 knockout mice have no detectable corin activity in the heart and develop salt-sensitive hypertension [37], indicating a key role of PCSK6 in regulating corin activity and blood pressure levels. Interestingly, PCSK6-mediated processing of corin is reduced in the presence of corin variants (T555I and Q568P) previously associated to hypertension and to heart disease in black people [39,40], as well as in the presence of corin variants (K317E, S472G, and R539C) previously identified in patients with preeclampsia and hypertension [41]. More recently, nine corin variants were identified in a Chinese population, of which eight were characterized for the first time. Among them, the p.Arg530Ser and p.Thr924Met variants had reduced proANP processing activity, due to endoplasmic reticulum retention and impaired PCSK6-mediated zymogen activation, respectively [42].

The proprotein processing enzyme furin is the mammalian prototype of the subtilisin-like serine endoprotease family. These enzymes possess cleavage specificity for sites involving multiple basic amino acid residues and process several precursor proteins belonging to a variety of regulatory peptides and proteins [43]. In fact, furin is the enzyme responsible for the cleavage of proBNP and proCNP. It also cleaves pro-renin receptor, epithelial sodium channel, and TGF-β [43]. Based on its functional properties, the gene encoding furin can be considered a suitable candidate for hypertension development. However, despite its involvement in several pathways, only weak evidence in favor of a role of furin gene in hypertension has been provided, so far, by both a GWA study and a few case–control association studies [44,45].

The deletion of *NPRA* gene leads to salt-sensitive hypertension and left ventricular hypertrophy (LVH) in mice [12]. An insertion–deletion mutation in the 5’-UTR region of the human *NPRA* gene has been described [46]. The mutant allele lacks eight nucleotides and alters binding sites for the AP2 and *zeste* transcriptional factors. The transcriptional activity of the deletion allele is shown to be 30% lower than that of the wild-type allele. Subjects carrying the deletion allele were significantly more common in a Japanese population of hypertensive patients compared to the normotensive group [46]. These findings suggested that the deletion of the *NPRA* reduces receptor activity in Japanese individuals and may confer increased susceptibility to developing essential hypertension or left ventricular hypertrophy.

More interesting findings were obtained for the *NPRC* gene encoding the NP clearance receptor. In fact, in two large studies, a gene-centric array and a GWA study identified few blood pressure loci, including one containing NPRC [44,47]. A small case-control association study, utilizing SNPs belonging to NPRC, detected an association with hypertension and even with family history of hypertension [48]. A recent study had the merit to confirm, first of all, using a genome-wide approach, the existence of two independent blood pressure-related signals within NPRC. Moreover, the biological relevance of one variant (rs1173771) was demonstrated by showing its association with lower NPRC gene and protein expression in vascular smooth muscle cells [49].

In contrast to the significant results obtained with both NPRA and NPRC, no relevant variants were identified in the *NPRB* gene in association with essential hypertension. 

## 4. Impact of NPs on Predisposition to Develop Hypertensive Target Organ Damage

The anti-remodeling properties of NPs may certainly explain their implication in the protection from the development of micro- and macrovascular damage, as well as of cardiac damage, that is associated with hypertension.

One of the most dramatic results of the hypertensive vascular damage is the occurrence of stroke [50]. The first evidence that *Nppa* was involved in the pathogenesis of stroke associated with hypertension was obtained in the animal model of the stroke-prone spontaneously hypertensive rat (SHRSP) through a genetic linkage analysis approach [51]. The latter was performed in a F2-hybrid cohort obtained from the SHRSP/stroke resistant SHR (SHRSR) intercross, considering the latency to stroke occurrence upon a high-salt dietary regimen as the stroke phenotype [51]. In fact, *Nppa* turned out to map, together with *Nppb*, at the peak of linkage of a quantitative trait locus (QTL) for stroke on rat chromosome 5 in this rat model [51]. The sequence analysis of both genes revealed the presence of two functional variants within *Nppa* (but none within *Nppb*), one located within the promoter region, the second one located within the coding part (exon 2). The *Nppa* variants were shown to cause an altered regulation, pro-peptide processing, and peptide function [52,53], further supporting their contribution to the development of cerebrovascular damage in SHRSP. Furthermore, the altered regulation of *Nppa* expression in the brain was found to co-segregate with increased susceptibility to stroke in the SHRSP/SHRSR F2-hybrid cohort [54].

Of note, the *Nppa* was excluded as a candidate gene for stroke in the SHRSP/Wistar Kyoto (WKY) genetic linkage analysis approach [55,56]. The different genetic background and the different stroke phenotype used in the latter study (the brain injury area produced by the middle cerebral artery occlusion, MCAO, as opposed to the stroke latency upon high salt diet used in the previous model) may certainly explain, at least in part, the different result. Despite the negative result of the genetic analysis in the study by Jeffs et al. [55], more recent evidence demonstrates that cerebral overexpression of ANP is vital to protection from brain damage in rats exposed to MCAO [57]. In fact, increased levels of ANP, achieved through an intraventricular injection, caused a significant reduction of the brain injury area [57]. Consistently it is noteworthy that a pharmacological treatment of high-salt fed SHRSP with a novel class of drugs (the type 1 angiotensin II receptor and neprilysin (NEP) inhibitor, ARNi) led to a dramatic reduction of stroke occurrence along with a significant increase of ANP levels in both circulation and brain tissue [58]. The latter evidence reinforces the beneficial cerebrovascular properties of ANP. 

Moreover, when translating the experimental findings to human disease, NPPA was confirmed as a risk factor for stroke in the presence of specific variants responsible for either altered gene expression or an altered protein function [26]. In particular, the T2238C variant, falling within exon 3 of the gene and causing the synthesis of a peptide carrying two extra arginines, is the most common NPPA variant associated with stroke [59,60]. The mutant peptide favors stroke through the production of an altered endothelial function, reduced endothelium-dependent vasodilation, vascular smooth muscle cells constriction, increased platelet aggregation through a NPRC deregulated activation [61,62,63,64,65,66,67]. Interestingly, subjects carrying this variant are also at increased risk of ischemic heart disease [66]. Finally, a gene-by-treatment interaction was observed for the T2238C NPPA variant and the use of diuretic drug (chlortalidone) in a large cohort of hypertensive patients from the ALLHAT trial [68]. In this study, patients carrying the C variant allele had a better response to the diuretic therapy [68], suggesting a defective diuretic action of the mutant peptide. 

The anti-hypertrophic and anti-fibrotic effects of ANP underlie its involvement in the pathogenesis of LVH in hypertension. The modulation of cardiac hypertrophy and fibrosis by ANP is achieved through the activation of several signaling pathways, such as the calcineurin/NFATt, sodium exchanger NHE-1, and TGFβ1/Smad pathways [69]. Thus, ANP behaves as a friend within the heart to protect it from stress stimuli [70]. At the experimental level the knockout of *Nppa* leads to cardiac hypertrophy in mice [10]. The same result is achieved when the *NPRA* gene is lacking [12] and also in the absence of corin [36]. In these models, the degree of LVH is unrelated to the levels of high blood pressure. In humans, we demonstrated that a promoter variant of *NPPA* causing reduced expression was associated with LVH in a cohort of hypertensive patients independently from the blood pressure levels [71]. Similarly, patients affected by the metabolic syndrome (that includes the diagnosis of hypertension) have plasma ANP levels inversely related to the degree of LVH such that the lower the ANP levels, the higher the degree of cardiac hypertrophy [72].

All components of the NP family are expressed in the kidney where they preserve renal function [73]. As a consequence, alteration of the NP system may contribute to renal damage occurrence in hypertension. In this regard, it has been reported that targeted disruption of NPRA provokes renal tubular damage and interstitial fibrosis with inflammation [74]. On the other hand, chronic ANP treatment ameliorates hypertension and end-organ damage in the kidney by reducing oxidative stress, increasing NO system activity, and diminishing collagen content and apoptosis in the SHR animal model [75]. Consistently, a significant protection from renal damage was also observed in the SHRSP receiving ARNi compared to SHR receiving valsartan alone [58]. As pointed out above, only ARNi treatment is able to increase the ANP levels.

NT-proBNP/BNP serves as a useful diagnostic and prognostic tool in heart failure, that is a frequent complication of the hypertensive cardiac damage. Of note, this component of the NP family has revealed a significant role into the prognosis of target organ damage in hypertensive patients [76,77]. However, in contrast to ANP, a direct involvement of BNP in the susceptibility to develop hypertensive target organ damage has not been proven either in animal models or in humans.

NPRC plays a fundamental role in NP clearance [2]. Specifically, it recognizes an 8 aa linear fragment of ANP molecule to perform peptide clearance. This process requires the ANP-NPRC internalization and is followed by ANP hydrolysis by lysosomes. In addition to its well-known function in NP clearance, NPRC can produce biological effects through the inhibition of adenylate cyclase and the involvement of Gi inhibitory proteins. Of note, its activation mediates the deleterious effects of the abovementioned T2238C ANP variant [62,63,64]. Evidence of a role of NPRC variants in the pathogenesis of hypertensive target organ damage, such as stroke, has been suggested by a GWA study [78]. We demonstrated that NPRC –55C > A transition contributes to the risk of early-onset ischemic stroke in an Italian cohort [79].

## 5. NP-Based Therapies for the Treatment of Hypertension and Related Organ Damage

This issue will be afforded by other articles from eminent scientists in this special issue of the Journal. As underscored by the experimental and human evidence discussed above, it is clear that within the NP family, ANP is the most suitable and efficacious anti-hypertensive agent. However, ANP cannot be administered orally. After many years of efforts, failures, and discoveries one later solution has been proposed in the form of a combination of a NEP inhibitor and an angiotensin II AT1 receptor blocker. NEP is responsible for the degradation of NPs, particularly ANP, and of angiotensin II [80]. This pharmacological combination, called ARNi, favors the protective effects of increased ANP levels along with inhibition of the negative effects of increased angiotensin II. ARNi is currently only indicated for the treatment of heart failure with reduced ejection fraction, but it shows potential to become a great solution for the treatment of hypertension, as documented by a few available experimental and human studies [58,81,82,83,84,85,86].

In summary, as represented in Figure 1 and Figure 2, the molecular implications of the NP system, in the pathogenesis of hypertension and related vascular damage, are of key relevance. This system offers an excellent example in the field of translational medicine. 

## Figures and Tables

**Figure 1 ijms-20-00798-f001:**
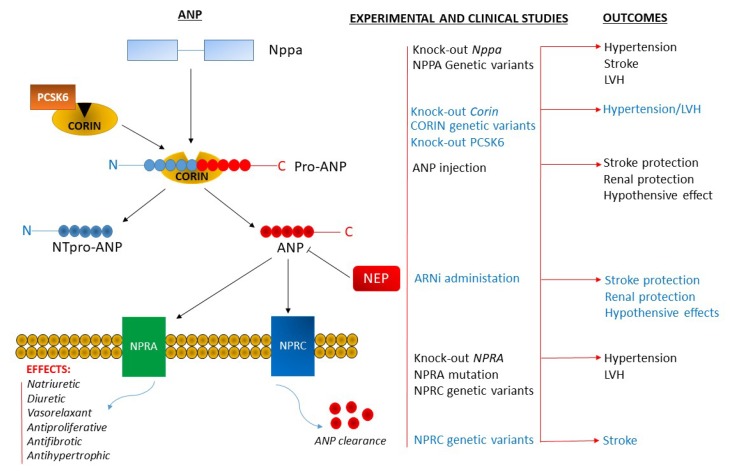
**Left side**: Schematic representation of the ANP processing, receptor interaction, and physiological functions. The ANP clearance by both NPRC and NEP is also shown. **Right side**: Summary of the experimental and clinical evidence showing the pathological consequences of the ANP, NPRA, corin gene deletions and gene mutations. ANP, atrial natriuretic peptide; ARNi, angiotensin type 1 receptor neprilysin inhibitor; LVH, left ventricular hypertrophy; NEP, neprilysin; NPPA, ANP gene; NPRA and NPRC, type A and type C natriuretic peptide receptors; NTpro-ANP, amino terminal pro-ANP; PCSK6, proprotein convertase subtilisin/kexin-6.

**Figure 2 ijms-20-00798-f002:**
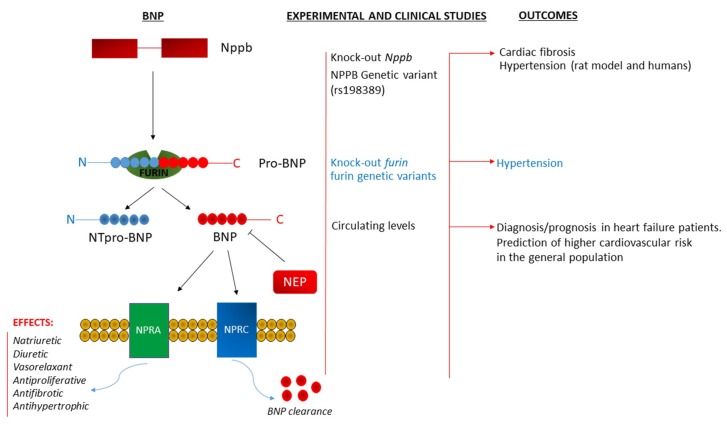
**Left side**: Schematic representation of the BNP processing, receptor interaction, clearance, and physiological functions. **Right side**: Summary of the experimental and clinical evidence showing the pathological consequences of the BNP and furin gene deletions; hBNP gene mutation and altered circulating BNP levels. BNP, brain natriuretic peptide; NEP, neprilysin; NPPB, BNP gene; NPRA and NPRC, type A and type C natriuretic peptide receptors; NT-proBNP, amino terminal pro-BNP.

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
