# Peer review of "Molecular Implications of Natriuretic Peptides in the Protection from Hypertension and Target Organ Damage Development"

_ijms, 2019, doi:10.3390/ijms20040798_

Round 1

Reviewer 1 Report

The review by Rubattu et al.  is well-written, properly organized and exhaustive. I have minor edits to suggest which are listed below:

Line 80: According to the study conducted by Arora et al (Journal of Clinical Investigation 2013) the variant rs5068 affects the “quantity” rather than the “stability” of the ANP transcript. The minor allele of rs5068 interferes with microRNA 425 binding to the NPPA 3’ UTR. In the presence of rs5068 minor allele, microRNA 425 cannot bind to NPPA and exert its inhibitory effect on ANP transcription.

Line 81: As properly done for the other studies on genetic variants cited in the manuscript, besides the review article by Rubattu et al (current reference 19) it would be appropriate to cite the original articles showing the association between rs5068 and blood pressure/hypertension: 1) Newton-Cheh et al, Nature Genetics 2009; 2) Cannone et al, Journal of American College of Cardiology 2011; 3) Cannone et al, Diabetes Care 2013; 4) Arora et al, Journal of Clinical Investigation 2013

Line 192 “T2238C….is the most commonly NPPA variant associated with stroke” : Besides the review article by Rubattu et al (current reference 19) the original studies showing an association between T2238C and stroke should be cited: 1) Rubattu et al, Stroke 2004; 2) Cannone et al, Hypertension 2013

Figure 1: Under “OUTCOMES” they list “Diagnosis/prognosis in CVDs. Prediction of higher cardiovascular risk in the general population”. In the manuscript, while the role of BNP as biomarker in heart failure is mentioned, circulating levels of ANP are discussed in relation to experimental models or genetic studies not as biomarker. I suggest that the above item is deleted from the “OUTCOMES” list in order to create continuity and consistency between the text and figure 1.

Author Response

We thank this Reviewer for his/her critical comments on our review article. We share the opinion of this Reviewer. In fact, there is no doubt that natriuretic peptides represent a fundamental mechanism of protection from hypertension and target organ damage. Their increase is protective whereas their decrease contributes to hypertension development. The knowledge that natriuretic peptides behave as an anti-hypertensive hormonal mechanism was proven many years ago, as we had already reported at the beginning of our article (pg.2, lines 47-53; refs. 4, 5).

On the other hand, the molecular genetic approach, that has been undertaken over the last 3 decades, has the merit to support the fundamental concept of natriuretic peptides as a protective mechanism toward hypertension. In fact, it has been proven that genetically induced decrease of either ANP or of its receptor are pro-hypertensive. Lack of the procorin processing enzyme, as well as of corin itself, lead to hypertension due to the ANP decrease. Consistently, ANP gene mutations leading to increased ANP expression lower blood pressure levels.

Therefore, the existence of molecular implications underlying the involvement of NPs in hypertension, while providing a deeper understanding of the phenomenon, does not go against the common view of NPs as a protective anti-hypertensive hormonal mechanism but it is fully consistent with it.

The present review article is focused on the discussion of the most relevant molecular aspects, that are widely documented in the literature, concerning the role of NPs, as well as of other components of the system, in hypertension in different animal models and in humans. These molecular aspects represent the basis of the anti-hypertensive effects of NPs and integrate the evidence about the role of modified circulating natriuretic peptides levels. We state this concept in the abstract (lines 15, 16) and in the text at pg. 2, lines 58, 59 and at pg. 3, lines 112, 113.

With regard to the protection from target organ damage in hypertension, the literature offers several insights on the molecular mechanisms underlying the anti-hypertrophic, anti-inflammatory, anti-fibrotic effects of NPs that contribute to their protective effects toward the development of cardiovascular damage. We state this concept in the abstract (lines 25, 26) and in the text at pg. 4, lines 167-169. This article discusses the most relevant mechanisms in order to provide a comprehensive review of the current knowledge.

Once again, the molecular genetic approach does not go against the concept of NPs as a mechanism of protection from development of target organ damage, but it rather explains it in deeper details.

According to the Reviewer criticisms, we modified the title of the article and we added his/her suggestions to the revised manuscript. We also introduced the evidence of the role of NPs in primary aldosteronism, in pheochromocytoma, in hyperthyroidism, as well as the effect of diuretics on NPs levels. See pg. 2, lines 53-57; new refs. 6-9.

Please note that all changes in the title and throughout the text have been highlighted in yellow.

Reviewer 2 Report

 The review reports the studies on a possible involvement of natriuretic peptides in the pathogenesis of hypertension

The report is of interest but the authors should stress out that natriuretic peptides are not involved in hypertension but rather their increase is a mechanism of protection from hypertension and the related organ damage.

ANP for example is increased in primary aldosteronism and probably is involved in the escape of the kidney to the action of aldosterone. NP are also increased in pheochromocytoma.

I suggest stressing this concept also in the title and over the manuscript. The possible protective mechanism of these substances and the possible relationship with catecholamines thyroid hormones. The action of NP si similar to the effectof diuretics. The authors should also report the effect of diuretics in reducing NP levels.

For example, in the abstract lines 21-.26 it seems that NPs contribute to the susceptibility to hypertensive target organ damage development.  I would write the opposite : NPa increase as a protective reaction to prohypertensive and profibrocic substances and for thgis reason they are increased in several type of hypertension.

Author Response

We thank this Reviewer for his/her critical and careful revision of our article and for the suggestions provided in order to improve it.

We modified accordingly the manuscript.

Previous line 80: we now better specify the findings from Arora et al. See pg. 2, lines 85-89; new ref. 23.

Previous line 81 (current line 89). We added the indicated refs. (see new refs. 24, 25).

Line 192 (current line 201). We added the indicated refs. (see new refs 59, 60)

Figure 1. As suggested, we removed from the outcomes the “Diagnosis/prognosis in CVDs. Prediction of cardiovascular risk in the general population.” See revised figure 1. Thank you.

Please note that, in order to accomplish the criticisms received from the Reviewer 1, we modified the title of the article and we introduced few changes throughout the manuscript that are highlighted in yellow.

Round 2

Reviewer 2 Report

the authors have answered to all the questions